# Towards Cross-Table Masked Pretraining for Web Data Mining

## ABSTRACT

Tabular data — also known as structured data — pervades the landscape of the World Wide Web, playing a foundational role in the digital architecture that underpins online information. Given the recent influence of large-scale pre-trained models like ChatGPT and SAM across various domains, exploring the application of pretraining techniques for mining tabular data on the web has emerged as a highly promising research direction. Indeed, there have been some recent works around this topic where most (if not all) of them are limited in the scope of a fixed-schema/single table. Due to the scale of the dataset and the parameter size of the prior models, we believe that we have not reached the "BERT moment" for the ubiquitous tabular data. The development on this line significantly lags behind the counterpart research domains such as natural language processing. In this work, we first identify the crucial research challenges behind tabular data pretraining, particularly overcoming the cross-table hurdle. As a pioneering endeavor, this work mainly (i)-contributes a high-quality real-world tabular dataset, (ii)-proposes an innovative, generic, and efficient cross-table pretraining framework, dubbed as CM2, where the core to it comprises a semantic-aware tabular neural network that uniformly encodes heterogeneous tables without much restriction and (iii)-introduces a novel pretraining objective — Prompt Masked Table Modeling (pMTM) — inspired from NLP but intricately tailored to scalable pretraining on tables. Our extensive experiments demonstrate CM2's state-of-the-art performance and validate that cross-table pretraining can enhance the performance of various downstream tasks.

## CCS CONCEPTS

• **Information systems** → *Data mining*; • **Computing methodologies** → *Artificial intelligence*.

## KEYWORDS

Data Mining, Pretraining, Tabular Data

**ACM Reference Format:**
Anonymous Author(s). 2024. Towards Cross-Table Masked Pretraining for Web Data Mining. In *Proceedings of Proceedings of the ACM Web Conference 2024 (WWW '24)*. ACM, New York, NY, USA, 11 pages. https://doi.org/XXXXXXX.XXXXXXX

## 1 INTRODUCTION

Since the launch of WWW or the Web, the internet industry has accumulated a tremendous amount of data, alongside the continual

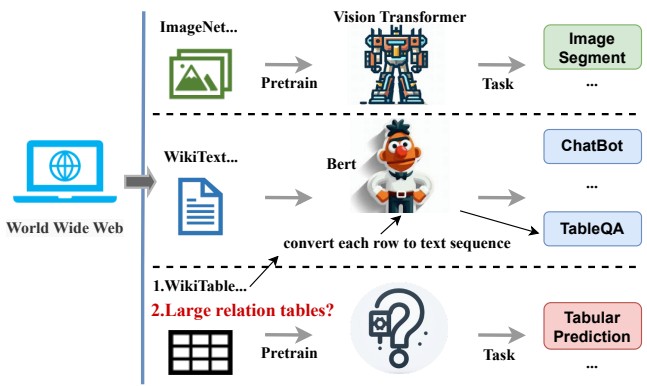

**Figure 1: Compared to the mature pretraining techniques in CV or NLP, how to pre-train a universal tabular model for mining widely prevalent relational tables on the Web remains an underexplored area. We focus on solving it.**

development of related infrastructural systems such as relational databases and others. In modern days, these data have become a pivotal pillar for the internet industry. It is widely acknowledged that properly mining the data can significantly profit advertisement campaigns, a recommendation system, risk management etc.

Indeed, as the data scales further and faster, the recent pretraining technique is very promising to better foster the mining process. Recently, the pretraining has notably "washed" the research terrain of natural language processing (NLP), computer vision (CV), and speech recognition [2, 23, 29], as shown in Figure 1. In spite of their superb successes, we argue that such a technique is still very much untapped for the tabular data (or the structured data) which is commonly extracted from relational database systems. Indeed, a limited number of prior works, such as TaBert [47], TCN [40] and Turl [9], have proposed to involve tabular data in the pretraining process. However, these solutions can arguably deemed as a combinatorial component in conjunction with the pretraining on natural language. In correspondence, their targeted tasks, such as TableQA [36], are also commonly deemed as a form of extension from the conventional QA task that the TableQA task constitutes the answers into a provided table. Also, the dimensions of the tabular data involved in the prior work are quite limited, averaging within 50*20 or less [17], which is much smaller by several order of magnitude than the scale of the Natural Language Processing pretraining. If resorting to the recipe from pretraining for NLP or CV, this scale of data is far insufficient. Despite all the above, in sharp contrast, the tabular data generated and stored from the Web is of much higher quantity than the other forms of data, owing to the unanimous utilization of databases and related software stacks.

On a separate but related line, a few recent works have made attempts to learn "contextualized representation" for the tabular data by neural network, such as [1, 16, 33], and proactively advocate the potential for pretraining. Despite the promise, most, if not all, of these work comply with their research within the scope

**Table 1: Existing cross-table learning efforts.**

| Methods
Properties | TransTab [43] | PTab [25] | XTab [51] | CM2 (ours) |
|---|---|---|---|---|
| Pretraining Scale (Tables) | <10 | <10 | 52 | >2000 |
| Unified Feature encoder | ✓ | ✓ | ✗ | ✓ |
| Regression Task | ✗ | ✗ | ✓ | ✓ |
| Classification Task | ✓ | ✓ | ✓ | ✓ |

of single-table training with pre-fixed schema meta. While this goal of learning contextualized representation partially aligns with pretraining, we emphasize that the key barrier hindering them from becoming impressive, general table pretraining models is limited capacity for cross-table learning. Few efforts have been made to explore this thus far, and all of them are included in Table 1. Regrettably, they merely conducted experiments on a small-scale toy dataset, coupled with less innovative approaches, unable to clearly validate cross-table migration of shareable knowledge. To summarize, despite the recent work around table pretraining, due to the limited scalability and the unresolved technical hurdles paving the way for cross-table pretraining, we believe we have not yet reached the "BERT [10] moment" for the ubiquitous tabular data.

## 1.1 Challenges

We characterize these three following challenges for tabular data pretraining.

**C1. Limited Pretraining Corpus.** While the available tabular data undoubtedly abounds on the Web, the current available open-sourced datasets are generally small and scattered around. This effectively caused the prior work to conduct their pretraining and validation experiments within the regime of less than 100 tables.

**C2. Unsuitable Tabular Data Encoder.** Unlike image or natural language data that generally express strong uniformity inherently, tabular data is notably more versatile. For instance, it may carry both continuous or categorical data, and compose variant column types and quantities depending on the schema meta-information varying across databases. However, current tabular encoders either struggle to uniformly encode heterogeneous tables, or they resemble table-to-text conversion models that learn contextual relationships at the *word level* without a deeper understanding of table structural information. Furthermore, previous works have overlooked the unique *permutation invariance* property of tabular data — as one can permute or switch any selected rows and/or columns, the table remains identical.

**C3. Non-universal Pretraining Objectives.** Coupling with a novel tabular encoder, a proper pretraining objective function (or a function set) — that supports the pre-trained model to learn the shareable knowledge or information across tables — is demanded. Previous studies often directly applied pretraining objectives from computer vision or natural language processing without considering the distinct characteristic of tabular data, namely the lack of clear contextual or spatial structure. This oversight results in these pretraining objectives being ill-suited for capturing the prior structural information inherent in tabular data. In other words, the current research lacks a tailored pretraining objective specifically designed for tabular data to learn the underlying relationships between columns.

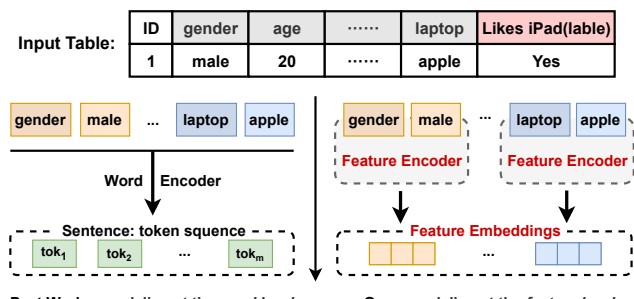

**Figure 2: The differences in combining column schema of table between past works (good for structured semantic understanding) and CM2 (better suited for tabular prediction).**

## 1.2 Our Solution

As a pioneering effort to enable large-scale cross-table pretraining, our endeavors encompass two key facets. In the correspondence to the major challenge C1 we mentioned, with this work we first devoted considerable effort to curate an extensive, high-quality tabular dataset (details in Section 3) sourced from the Web, serving as a foundation for table pretraining experiments of this work as well as the future advancement for the field. The second aspect of this work is a novel, generic and efficient **C**ross-table **M**ased **M**odeling (**CM2**[1]) pretraining framework as we devise, that is aimed to solve the challenge C2 and C3.

To be slightly more specific, towards C2, we propose an innovative semantic-aware tabular neural network. Considering that the atomic/basic unit in tabular data is the feature column, the first step in a tabular neural network is usually to encoder each feature values to a low-dimensional vector representation, similar to word embedding [28] in NLP. Previous work has typically built an embedding lookup table for each feature values. However, considering cross-table scenarios, such a mapping table is infinite, clearly infeasible, and not conducive to knowledge transfer across different tables. In CM2, we design a semantic-aware tabular data encoder. By purposely designing a fusion module in the representation space that integrates the information from both the cell and its corresponding schema information, we manage to cope with the heterogeneous tables uniformly. As compared with the very few related methods (illustrated in Figure 2) that often uncontrollably merge information from the cells and schema altogether, this means of the encoding semantically informs the model of the information from the finest-grained level. A similar method was proposed in NLP [21]. We also tuned the Transformer [39] architecture to adapt it to the permutation invariance property of the tabular data.

For the challenge C3, we devise a novel objective, dubbed as prompted Masked Table Modeling (pMTM). That is, inspired by the MLM objective used in BERT training [10], we mask the data of a randomly picked sub-table and then let the model predict the missing values. Notably, we posit that simply copying the MLM objective would not suffice due to the insufficient information implied for table data recovery. We therefore consummate the objective by providing a cule—pointing to the corresponding schema information—which can also be deemed as a form of *prompt*. And we

---

[1]Source code can be found at https://anonymous.4open.science/r/CM2-1F5F.

believe that if the model can predict the masked features from the retained features, then the model can learn the underlying relationship between the features. Similar to CV or NLP, this relationship serves as the foundation to manifest the shareable knowledge that is transferred across tables. In addition, we also validate some other pretraining objectives (Experiment 5.5.2), such as vanilla supervised transfer learning and contrastive learning, which have been fully explored for tabular data in previous works [38, 43].

Last but not least, we have released a large pre-trained tabular model $CM2_{V_1}$[2] (approximately 50 million parameters) trained on 2k datasets, which supports fine-tuning or few-shot learning for prediction on tables without constraints. It is true that the number of parameters of CM2 still falls behind the recent large language models, such as LLaMA [37] (7, 13, 33 and 65 billion) or ChatGPT [29] (175 billion). But just as mentioned, we hope to use CM2 to establish the "BERT-moment" for table pretraining, where CM2 roughly reaches close to the BERT's parameters volume (i.e. BERT-base 110 million). We believe this direction is indispensable, because of the dominant existence and storage of tabular data on the Web.

## 2 RELATED WORKS

In this section, we present and analyze three directions around downstream tasks, pretraining methods, and pretraining corpus on tabular data to explain why the "BERT moment" has not yet been achieved in the tabular data domain.

**Tabular Downstream Tasks:** The diverse approaches to modeling tabular data can be classified based on their relevance to downstream tasks, including: **(1)**-*tableQA* [14, 47], which answers the questions based on the table; **(2)**-*table to text* [26, 30], where textual descriptions are generated from a given table; **(3)**-*table interpretation* [9, 18], which aims to uncover semantic attributes in tabular data, such as entity linking, and so on. **(4)**-*Tabular data generation*, which is applicable to privacy protection scenarios [46], enhances the performance of downstream models through data augmentation [4, 49], etc.

The above methods are essentially related to text understanding in NLP, and modeling is relatively simple and lacks a deeper understanding of structural information in tabular data. In contrast to these tasks, our work mainly focuses on more challenging *tabular prediction tasks*, suitable for a wide range of application scenarios, such as product recommendations [50], click-through rate predictions [34], etc. At the same time, we hope that our pre-trained model CM2 to be capable of supporting common real-world tasks involving tabular data, such as *anomaly detection* [6] and *missing value imputation* [35].

**Tabular Pretraining Methods:** In the domain of tabular tasks related to our research, certain studies have attempted to extend the success of pretraining then fine-tuning paradigm [1, 16, 33, 38, 48]. These approaches typically use contrastive learning similar to that in SimCLR [8] or reconstruct damaged table inputs. However, they are all limited to pretraining on a fixed-schema/single table. Recently, several studies have explored cross-table knowledge transfer. TabPFN [15] conducts pretraining of a prior model on *synthetic* dataset and exhibits promising results on small numerical classification task. Different from XTab [51], which uses data-specific

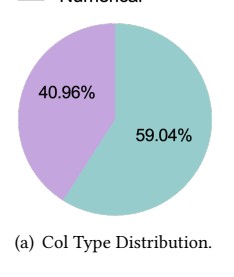

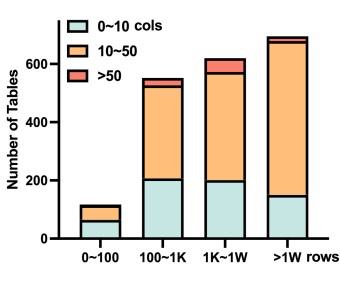

(a) Col Type Distribution.    (b) Dimensions of Tables.

**Figure 3: Statistics of our OpenTabs dataset composition.**

feature processors leading to challenges in knowledge transfer and model pretraining memory is disastrous as the number of datasets increases, we use a unified feature encoder. TransTab[43] and PTab [25] both adopt the table-to-text modeling approach and pretrain on a few relevant tables. All these efforts are either confined to a single table or suffer from a limited scale of pretraining corpus, thereby constraining the generalization capacity.

**Tabular Pretraining Corpus:** In recent years, there has been relatively little research on cross-table pretraining. We believe that a major obstacle in this field is the lack of large-scale and high-quality table datasets. Currently, there are available open-source table datasets, such as Wikitable [5] and GitTables [17], which are mainly used for table-based text understanding tasks (e.g., TableQA [36]). Due to the notably small scale of these tables, they are inadequate for learning deeper structural information within tabular data. Based on this, we contribute a large-scale tabular dataset OpenTabs in section 3.

## 3 OPENTABS: A LARGE-SCALE TABULAR DATASET FROM WEB

As discussed before, we need a large-scale, high-quality dataset to pretrain a universal tabular model. Just as many high-quality text datasets [10] significantly accelerated progress in NLP research, a similar catalyst is required to bring tabular data to the "BERT moment". In this work, we contribute a substantial tabular dataset OpenTabs, sourced primarily from public websites like OpenML[3], UCI[4], Kaggle[5], and CATALOG[6]. Note that, as the quality of the tabular data has a significant impact on the pre-trained model, we have invested a considerable amount of effort in thorough data filtering and cleaning procedures to maintain dataset quality. Please see Appendix C.1 for more details. And some statistical information about this dataset is shown in Figure 3. At present, OpenTabs hosts an extensive collection of large-scale tables, including approximately 46 million tabular samples. **We have open-sourced OpenTabs[7] in response to challenge C1 and hope to facilitate future research in the field of table pretraining, particularly towards the cross-table learning.**

---

[2]$CM2_{V_1}$

[3]https://www.openml.org/
[4]https://archive.ics.uci.edu/datasets
[5]https://www.kaggle.com/
[6]https://catalog.data.gov/dataset
[7]OpenTabs

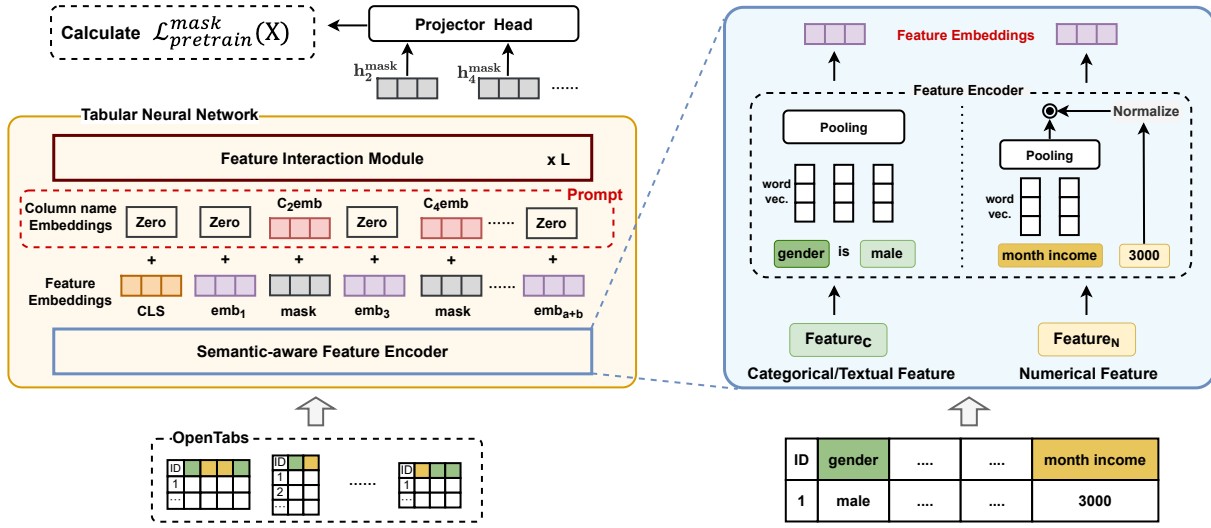

Figure 4: The overview of the proposed cross-table pretraining framework CM2. OpenTabs (Section 3) is the pretraining dataset contributed by us. Firstly, we employ a feature encoder to uniformly process these heterogeneous tables and obtain feature embeddings. Then we mask some features and replace them with a shared, learnable vector. Finally, our pretraining objective aims at recovering these masked features based on the retained features and schema information prompt.

## 4 METHODS

In this section, we describe our cross-table pretraining framework CM2 (overview in Figure 4) in depth. We first introduce a semantic-aware tabular neural network as an innovative tabular data encoder to resolve challenge C2 in Section 4.2. Then we detail proposed cross-table pretraining objective pMTM to address challenge C3 in Section 4.3. Finally, we discuss how to fine-tune on downstream tasks in Section 4.4.

### 4.1 Problem Formulation

For a given tabular data $D = (\mathbf{x}_i, y_i)_{i=1}^n$ where $n$ refers to the number of samples. $\mathbf{x}_i = \{\mathbf{x}_i^{cat}, \mathbf{x}_i^{num}\}$ where $\mathbf{x}_i^{cat} = \{x_i^1, x_i^2, \ldots, x_i^a\}$ denotes all $a$ categorical/textual features, and $\mathbf{x}_i^{num} \in \mathbb{R}^b$ denotes all $b$ numerical features. $y_i \in \{1, 2, \ldots, T\}$ where $T$ refers to the total classes of labels. All samples share the same table header descriptions (column names) $\mathbf{C} = \{c^1, c^2, \ldots, c^{a+b}\}$. Our goal is to find the best possible prediction function $f_\theta$ to model the mapping between features and labels:

$$f_\theta(\mathbf{x}_i; \mathbf{C}) = y_i, \tag{1}$$

where $\theta$ refers to all trainable parameters of the function $f$.

### 4.2 Semantic-aware Tabular Neural Network

This section will introduce the novel tabular neural network that we have designed: a semantic-aware feature encoder followed by a feature interaction model to resolve the *heterogeneous tables* encoding and *permutation invariance* problems in **challenge C2**.

*4.2.1 Feature Encoder for Heterogeneous Tables.* Diverse tabular data typically exhibits distinct combinations of column types depending on the schema meta-information varying across databases. Therefore, prior works [12, 51] often employ data-specific

feature processors, or more specifically, they would establish an embedding lookup table for each feature. This greatly impedes these models from conducting cross-table learning.

In CM2, we propose a *semantic-aware feature encoder* to uniformly handle variable tabular column types and encode each feature into a low-dimensional vector representation. In particular, we analyze that the table is essentially a multimodal structured data, which contains both text (e.g., column names and categorical/textual features) and continuous values (e.g., numerical features). Based on this observation, first, we refine the atomic units "feature" within the table into a sequence of words, which also includes the corresponding column name schema information. This can be abstractly considered as a feature phrase and processed using the NLP approach. Second, we fuse the information of these word vectors to get the final feature embedding. This *semantic-aware feature encoder* has following two advantages:

**It can uniformly accept inputs from heterogeneous tables**, which serves as a necessary condition for enabling cross-table pretraining. Thanks to this ingenious design, we no longer need to feature embeddings lookup table for each data. While we transform challenging feature embedding in cross-table scenarios into unified and limited word embedding.

**The shareable knowledge can be maximized to properly transfer across different tables by semantic information in the schema.** For example, the feature values "hot" and "high" can share cross-table knowledge through the semantic similarity of "temperature is hot" and "temperature is high". The "apple" can be distinguished by combining the column names "fruit is apple" and "my laptop is apple" to prevent unentangleable confusion in cross-table learning. In contrast to prior table-to-text approaches [25, 43, 47], our tabular neural network learns at *feature level*, whereas the former models at *word level*. Our approach can capture deeper structural information in tabular data, particularly for large-scale

tables. Empirical experimental results (Section 5.3) demonstrate that we achieve better performance on more challenging and general tabular prediction tasks. The right part of Figure 4 illustrates the details about how we handle the categorical/textual and numerical features separately:

$$e^{j \in cat} = Encoder(x^j) = Pooling(Tokenize(Concat(c^j, x^j))) \quad (2)$$

$$e^{j \in num} = Encoder(x^j) = Pooling(Tokenize(c^j)) * x^j \quad (3)$$

$$H^0 = Encoder(x^{cat}, x^{num}) = [e^0, e^1, e^2, ..., e^{a+b}] \quad (4)$$

where $H^0$ represents all feature embeddings and serves as the initial input for the subsequent feature interaction module. We use a pre-trained BERT [10] model, which contains generic semantic knowledge, to *tokenize* and generate the corresponding embedding for each token. In Experiment 5.3 we discussed different *pooling* strategies. Please note that we normalize numerical features here to avoid confusion during cross-table pretraining. Because the same numerical feature may have different measurement units across different tables (e.g., meter vs. centimeter).

### 4.2.2 Feature Interaction.
As discussed before in challenge C2, our feature interaction model needs to be capable of handling the *permutation-invariant* natures of tabular data. Additionally, to facilitate cross-table learning, our model must accommodate tables with varying column lengths. Taking all the above into account, we employ a transformer [39] with a multi-head attention mechanism to model feature interactions:

$$MultiHead(\mathbf{H}^l) = Concat(head_1, ..., head_i, ..., head_h))\mathbf{W}^O, \quad (5)$$

$$head_i = Attention(\mathbf{H}^l\mathbf{W}_i^Q, \mathbf{H}^l\mathbf{W}_i^K, \mathbf{H}^l\mathbf{W}_i^V), \quad (6)$$

$$Attention(\mathbf{Q}, \mathbf{K}, \mathbf{V}) = Softmax(\frac{\mathbf{Q}\mathbf{K}^T}{\sqrt{d}})\mathbf{V}, \quad (7)$$

where $\mathbf{H}^l \in \mathbb{R}^{n \times d}$ is the input of the l-th layer; $\mathbf{W}^O \in \mathbb{R}^{d \times d}$ is parameter matrix; $\mathbf{W}_i^Q$, $\mathbf{W}_i^K$ and $\mathbf{W}_i^V \in \mathbb{R}^{d \times d_{head}}$. $d_{head} = \frac{d}{h}$ is the dimension of each attention head. We discard *positional encoding* to accommodate the *permutation invariance* of tabular data. Notably, we abstain from considering convolutional or recurrent neural network families because these models incorporate spatial/sequential positions as part of their inductive biases. Inspired by BERT [10], we add a special classification token ($\mathbf{e}^{CLS} \in \mathbb{R}^d$), which is expected to aggregate all feature information and serve as a sample representation, at the first position of the input sequence in each layer. Finally, through the projection head, we obtain high-order feature interaction outcomes $\mathbf{Z} = \{\mathbf{z}^{CLS}, \mathbf{z}^1, \mathbf{z}^2, ..., \mathbf{z}^{a+b}\}$.

## 4.3 Cross-table Pretraining Objective

Tabular data, owing to its heterogeneity, lacks a clear contextual or spatial structure, which can be distinctly observed in textual and image data as evident prior information. Consequently, the previous approach of converting tables into text sequences for learning contextual relationships in pretraining objectives is not suitable for our feature-level learning [9, 47].

In reality, there exist interdependencies among different columns in tabular data. For instance, the "income" feature is likely contingent upon both "job" and "company" features. Motivated by these observations and drawing inspiration from the MLM objective in BERT [10], we propose an innovative *prompt Masked Table Modeling* (pMTM) self-supervised cross-table pretraining task. For each tabular sample, we mask a certain proportion of features and then predict them based on the retained features and schema information *prompt*. If successful reconstruction is possible, we believe that the model has learned the underlying relationships between feature columns, which serves as shareable knowledge across tables. Different from previous tabular reconstruction endeavors [1, 48], **our first attempt to use column names as *prompt* to assist in predicting masked features.** For two purposes: **(i)**-Tabular data lacks general positional encoding due to *permutation invariance*, which is theoretically necessary for the recovery of masked data. Column name encoding replaces this role. **(ii)**-Facilitating cross-table knowledge transfer by the semantics embedded in column names. Furthermore, with a measure of self-promotion, we aspire to formally introduce the concept of "Masked Table Modeling" within this work, analogous to Masked Language/Image Modeling task [10, 45] in the NLP/CV domain. **Our aim is to inspire more researchers to explore masking strategies for tabular data.**

The left part of Figure 4 shows the overview of our pMTM pretraining method, which involves three main steps. First, in the input processor, we randomly mask approximately $p^{mask}$ features for each row ($p^{mask}$ is set to 35% in our experiments, and further ablation results are shown in Section 5.5.3). Second, we replace the masked features with a shared, learnable vector $\mathbf{e}^{mask} \in \mathbb{R}^d$, which is also called *mask token*. Here, importantly, we will add corresponding column name encoding for each *mask token*. Lastly, we feed these mask tokens and retained features into the L-layer transformer encoder to learn, and then reconstruct them. For the masked numerical features, we calculate the mean square error loss with the original feature values $x_i$. For the masked categorical features, we compute the cosine similarity with the original cell value embedding $\hat{\mathbf{e}} \in \mathbb{R}^d$, which is obtained in the same way as Section 4.2.1 but without the column names. pMTM loss as follows:

$$\mathcal{L}_{pretrain}^{mask}(\mathbf{X}) = \frac{1}{|B|}\sum_{i \in B}\Phi(\mathbf{x_i}, \hat{\mathbf{e}_i}, \mathbf{z_i}), \quad (8)$$

$$\Phi(\mathbf{x_i}, \hat{\mathbf{e}_i}, \mathbf{z_i}) = \frac{1}{N^{num}}\sum_{j=1}^{N^{num}}(x_i^j - z_i^j)^2 + \frac{1}{N^{cat}}\sum_{j'=1}^{N^{cat}}(1 - sim(\hat{\mathbf{e}}_i^{j'}, \mathbf{z}_i^{j'})) \quad (9)$$

where B is the set of samples in a batch; $N^{num}/N^{cat}$ refers to the number of numerical/categorical(textual) features. We do not employ cross-entropy loss for categorical features because the same class in similar columns may be inconsistently labeled across different tables, which could lead to confusion when cross-table pretraining.

## 4.4 Fine-Tuning on Downstream Tabular Tasks

After cross-table pretraining, we discarded the original projection header, added a new task layer, and subsequently fine-tuned the parameters on downstream tabular tasks.

**Table 2: Comparison of CM2 with other shallow and neural network-based tabular modeling methods. CM2 is initially pretrained on OpenTabs (Section 3) and subsequently fine-tuned on these downstream tasks. Bold indicates the best results.**

| Dataset | Shallow Methods | | | Neural Network-Based Methods | | | | | | | | |
|---|---|---|---|---|---|---|---|---|---|---|---|---|
| | LR | XGBoost | LightGBM | DCN-v2 | AutoInt | MLP | FT-Trans | Saint | TabNet | TransTab | XTab | **CM2**(ours) |
| Breast | 0.9947 | 0.9429 | 0.9568 | 0.9627 | 0.9866 | 0.9710 | 0.9927 | 0.9894 | 0.9943 | 0.9941 | 0.9840 | **0.9961** |
| Cmc | 0.6935 | 0.6762 | 0.7038 | 0.6762 | 0.7041 | 0.6558 | 0.7267 | 0.7235 | 0.6722 | 0.7274 | 0.6915 | **0.7306** |
| Diabetes | 0.8263 | 0.6915 | 0.6932 | 0.7518 | 0.8144 | 0.7943 | 0.8266 | 0.8146 | 0.8110 | 0.8357 | 0.8094 | **0.8369** |
| Vehicle | 0.8912 | 0.9218 | 0.9153 | 0.9125 | 0.8883 | 0.8700 | 0.9136 | 0.8918 | 0.8051 | 0.9138 | 0.9109 | **0.9237** |
| Satimage | 0.9722 | 0.9893 | 0.9889 | 0.8023 | 0.953 | 0.9827 | 0.9831 | 0.9870 | 0.9831 | 0.9849 | 0.9864 | **0.9894** |
| Sick | 0.9255 | 0.9191 | 0.9267 | 0.8895 | 0.9379 | 0.6518 | 0.9512 | 0.9863 | 0.9806 | 0.9542 | 0.9718 | **0.9944** |
| Analcatdata | 0.5609 | 0.5253 | 0.5517 | 0.5563 | 0.5356 | 0.5590 | **0.5652** | 0.5371 | 0.5290 | 0.5639 | 0.5322 | 0.5470 |
| Pc1 | 0.7609 | 0.6379 | 0.6509 | 0.7965 | 0.7864 | 0.6169 | 0.7401 | 0.7483 | 0.7609 | 0.7924 | **0.8227** | 0.7764 |
| Adult | 0.8360 | 0.7891 | 0.7913 | 0.8923 | 0.8879 | 0.8995 | 0.9157 | 0.9151 | 0.9003 | 0.9139 | 0.9143 | **0.9158** |
| Phishing | 0.9786 | 0.9685 | 0.9577 | 0.9389 | 0.9789 | 0.9935 | 0.9940 | 0.9943 | 0.9903 | 0.8292 | 0.9915 | **0.9950** |
| Cylinder-bands | 0.7498 | 0.7659 | 0.7904 | 0.7465 | 0.7203 | 0.6304 | 0.8248 | 0.7642 | 0.5861 | 0.8201 | 0.6782 | **0.8377** |
| MiceProtein | 0.9973 | 0.9988 | 0.9998 | 0.8894 | 0.9112 | 0.9970 | 0.9982 | **0.9998** | 0.9404 | 0.9963 | 0.9991 | 0.9989 |
| Car | 0.7393 | 0.9974 | 0.9983 | 0.9896 | 0.9664 | 0.8539 | 0.9976 | 0.7806 | 0.9709 | 0.9005 | 0.9932 | **0.9999** |
| Segment | 0.9703 | 0.9939 | **0.9951** | 0.9746 | 0.9881 | 0.9796 | 0.9894 | 0.9868 | 0.9889 | 0.9893 | 0.9879 | 0.9934 |
| Porto-seguro | 0.4869 | 0.5000 | 0.5000 | 0.5162 | 0.5491 | 0.5177 | 0.5008 | 0.5814 | 0.5106 | 0.4803 | 0.5430 | **0.5931** |
| Amazon | 0.5315 | 0.5516 | 0.5233 | 0.5564 | 0.5372 | 0.5338 | 0.5358 | 0.5083 | 0.5190 | 0.5321 | 0.5606 | **0.6464** |
| mean | 0.8072 | 0.8043 | 0.8089 | 0.8032 | 0.8216 | 0.7817 | 0.8410 | 0.8255 | 0.8089 | 0.8267 | 0.8360 | **0.8609** |

## 5 EXPERIMENTS

In this section, we conducted extensive experiments to demonstrate the effectiveness and superiority of CM2.

### 5.1 Experimental Setup

*5.1.1 **Datasets**.* The experimental datasets consist of two parts:
**Upstream cross-table pretraining datasets.** We contribute a high-quality tabular dataset OpenTabs (Section 3), which includes a vast number of large-scale tables with rich semantic information in column names and performed strict data cleaning. We have open-sourced it in anticipation of advancing the tabular pretraining.
**Downstream tabular datasets.** We use 16 diverse public benchmark tabular datasets to evaluate the effectiveness of CM2. These datasets are from OpenML, UCI repository, and Kaggle and contain both binary and multi-class classification tasks. Additionally, extra data is used to validate our framework across diverse downstream tasks, including regression and anomaly detection; refer to the corresponding section for details. To ensure a fair and unbiased evaluation, we intentionally excluded these downstream tabular tasks from the pretraining corpus. We included the source of each dataset in the Appendix C.2.

*5.1.2 **Competing Methods**.* In order to assess the superiority of CM2, we conducted a comprehensive comparative analysis with various tabular modeling baselines: **(i)-Traditional shallow methods:** *Logitic Regression* [44], *Xgboost* [7], *LightGBM* [20]. **(ii)-Neural network-based methods:** *DCN-v2* [42], *AutoInt* [34], *MLP (Multilayer Perceptron)* [11], *FT-Transformer* [12], *TabNet* [1], *SAINT* [33], *TransTab* [43], *XTab* [51]. For detailed descriptions and implementations of these baselines, please refer to Appendix B.

*5.1.3 **Metrics**.* We follow previous work [12, 43] using AUC [27] as the main evaluation metric for the classification task. In addition, to verify the versatility of MC2 across various tabular downstream

tasks, we employed root mean square error (RMSE) and F1 score for regression and anomaly detection tasks, respectively.

*5.1.4 **Implementation Details**.* We report the performance of the pretrained models in all experiments. Other than in Section 5.3, where we compare CM2 with the model trained from scratch (w/o pretraining) to validate the benefits of pretraining. In the data pre-processing phase, we scale numerical features to [0, 1] by min-max normalization in all methods. For categorical features, we use ordinal encoding to represent them in the regular baseline methods. However, note that in our CM2, we use the raw textual values of the categorical features in order to better exploit their semantic information. We use a pre-trained BERT-base-uncased [10] model on Hugging Face[8] to obtain token embeddings that are rich in semantic information. We use the DeepSpeed [31] framework for parallel computation acceleration to improve the pretraining efficiency. For the sake of equitable comparison, consistent data partitioning and evaluation procedures are maintained for CM2 and all baselines, unless otherwise specified. All results are obtained by 5-fold cross-validation. Within each fold of the training set, we partition 20% as a validation set, which was utilized for hyperparameter selection and early stopping. See Appendix B for hyperparameters settings of all experiments. We also report the sensitivity analysis experiment of hyperparameters in Appendix 5.5.4.

### 5.2 Overall Performance

In this section, we report the overall performance of CM2. The results are shown in Table 2. CM2 achieves the *state-of-the-art* average performance, with **12** out of 16 datasets in total performing better than all baseline methods. On one hand, although the FT-transformer [12] currently stands as the most advanced method in

---

[8]https://github.com/huggingface

**Table 3: Ablation studies of feature-level learning and different pooling strategies.**

| Methods | CM2 Avg. result | Dataset Case | | |
|---|---|---|---|---|
| | | Phishing | Car | Porto-seguro |
| *word-level learning* | | | | |
| No-Pooing | 0.8342 | 0.8302 | 0.9039 | 0.4910 |
| *feature-level learning (different pooling strategies)* | | | | |
| Max | -0.0087 | -0.0008 | -0.0006 | -0.0154 |
| Average | **0.8609** | **0.9950** | **0.9999** | **0.5931** |
| Self-Attention | -0.0108 | -0.0091 | -0.0003 | -0.0033 |

fixed-schema/signal table setup, CM2 outperforms it by a significant margin. Therefore, we have reason to believe that cross-table pretraining is a potential direction and could enhance the model's capabilities by learning from diverse upstream datasets. On the other hand, compared to methods TransTab [43] and XTab [51] that similarly introduce the concept of cross-table learning, CM2 continues to hold substantial advantages. This indicates the superiority of the semantic-aware tabular neural network we design, as well as the prompt Masked Table Modeling (pMTM) tabular pretraining objective that we propose.

In summary, CM2 is the pioneering effort in the realm of large-scale cross-table pretraining. Our empirical experimental results validate the feasibility that learning shareable knowledge across different tables could enhance the model's generalization performance on diverse downstream tasks.

## 5.3 Ablation Studies

**Comparison with Learning from Scratch.** To further demonstrate the effectiveness of *cross-table pretraining*, we compare CM2 with learning from scratch (*w/o pretraining*). The results are shown in Figure 5. We observe that pre-trained model generally exhibit faster convergence rates and yield better results. This indicates that CM2 has learned beneficial shareable knowledge for downstream tasks through cross-table pretraining on large-scale tabular datasets. Therefore, **we have reason to believe that the proposed prompt Masked Table Modeling, as a tabular pretraining objective, can effectively capture the prior structural information inherent in tabular data** and help enhance the model's generalization capability across various downstream tasks.

**Feature-level Learning.** Previous work [25, 43] introduce the concept of table-to-text into tabular modeling, that learns contextual relationships at the *word level*. To demonstrate that maintaining *feature-level* learning is more suitable for capturing deeper structural information in tabular data, we conducted ablation experiments on more challenging tabular prediction task. Specifically, we do not pool all word embeddings into one feature embedding but feed them directly into the feature interaction model for learning, and keep the other experimental settings identical. Table 3 shows the results. We observed that *feature-level* learning indeed exhibits superior average performance and shows notable advantages (**+10%**) in certain dataset cases. Additionally, we further evaluate different pooling strategies, including average pooling, max pooling, and self-attention [39] pooling. The results are shown in Table 3. Among these strategies, the approach of average pooling stands

**Table 4: Performance of CM2 on regression, anomaly detection and missing value imputation tasks.**

| | *Regression Task* | | | | | |
|---|---|---|---|---|---|---|
| Methods | Elevators | Yprop | Topo | SAT11 | Diamonds | House_sales |
| XGB | 0.3477 | 1.0021 | 1.0319 | 0.5067 | 0.1411 | 0.3507 |
| MLP | 0.3392 | 0.9945 | 0.9855 | 0.6086 | 0.1965 | 0.3620 |
| FT-trans | 0.2956 | 0.9770 | **0.9661** | 0.6315 | 0.1456 | 0.3438 |
| XTab | 0.2959 | 0.9723 | 0.9755 | 0.5641 | 0.1401 | 0.3386 |
| **CM2**(ours) | **0.2924** | **0.9706** | 0.9778 | **0.4800** | **0.1391** | **0.3356** |

| | *Anomaly Detection Task* | | | | | |
|---|---|---|---|---|---|---|
| | Wine | | Vertebral | | Ionosphere | |
| Methods | F1 | Auc | F1 | Auc | F1 | Auc |
| ADTICL [32] | 0.7000 | 0.8864 | 0.1333 | 0.3990 | **0.9365** | **0.9798** |
| **CM2**(ours) | **0.8000** | **0.9864** | **0.5667** | **0.7904** | 0.9286 | 0.9713 |

| | *Missing Value Imputation Task* | | | | | |
|---|---|---|---|---|---|---|
| Methods | Cmc | Diabetes | Sick | Adult | Amazon | Breast |
| M_Ori | 0.6825 | 0.6754 | 0.9132 | 0.7860 | 0.5139 | 0.9541 |
| HyperImpute [19] | 0.6870 | **0.6823** | **0.9203** | 0.7731 | 0.5074 | **0.9599** |
| **CM2**(ours) | **0.6913** | 0.6791 | 0.9180 | **0.7895** | **0.5146** | 0.9521 |

out as it efficiently simplifies information extraction, ultimately leading to the best results.

## 5.4 Supporting Diverse Tabular Tasks

To validate that the tabular representations learned by CM2 can support various downstream tasks, similar to BERT's generalization across NLP tasks, we conducted additional tests in diverse scenarios. Previous works typically address singular task.

**Regression Problem.** To validate the extension of CM2 to regression scenarios, we have added a new task layer to predict continuous values. The details and source of each dataset can be found in Appendix C.2. Following the previous work [12], we standardize the target column. As can be seen in Table 4, CM2 has an excellent advantage on regression tasks, with **5** out of 6 diverse downstream tabular tasks performing better. This proves that CM2 is still applicable to regression scenarios.

**Anomaly Detection.** As a one-class classification problem, anomaly detection has many important application scenarios including tabular data. Its objective is to identify out-of-class samples in tabular data. In this study, we validate the effectiveness of CM2 in the context of anomaly detection. Specifically, building upon prior research [32], we replaced the backbone with CM2. The results are presented in Table 4, CM2 typically exhibits better performance. As a tabular data encoder, CM2 serves the task of anomaly detection by providing good representations of tabular samples.

**Missing Value Imputation.** Tabular data often come with impurity issues such as missing values. We attempted to use the CM2 to achieve missing value imputation. We randomly selected 6 datasets and followed previous work [19] to generate missing values using the *missing at random* (MAR) mechanism. For categorical features, we calculated the cosine similarity between the reconstructed features and other features in the missing feature column, and then selected the feature value with the highest matching degree as the filling; For numerical features, no processing is done. The experimental results are shown in Table 4, where "M_Ori" means training with the original data processed by the MAR, and the evaluation model is LightGBM. We found that CM2 performs well in the missing value imputation scenario.

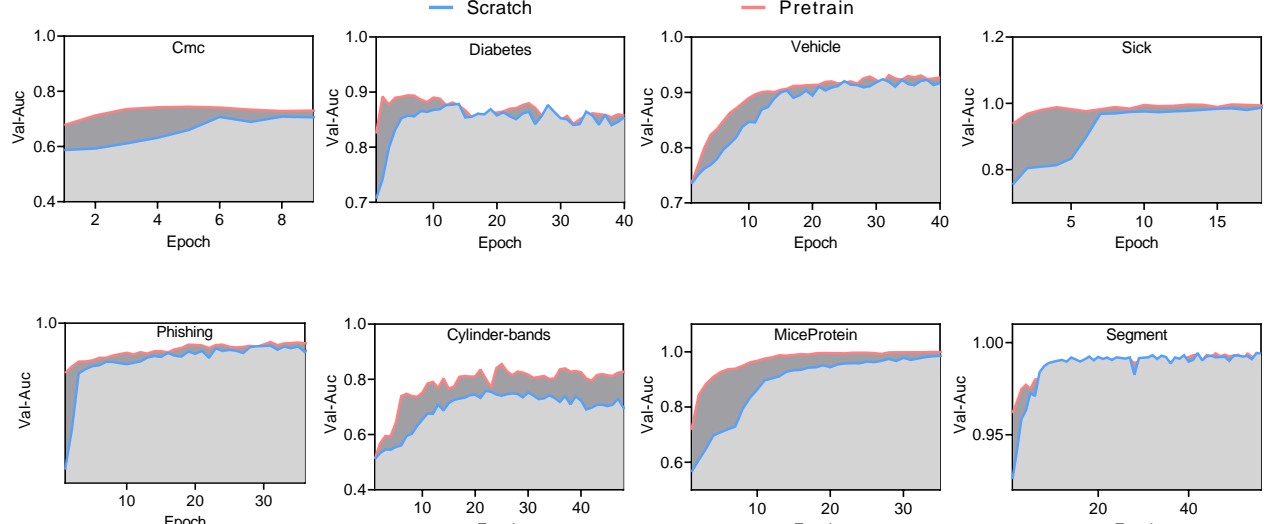

Figure 5: Ablation studies on comparing CM2 with learning from scratch (*w/o pretraining*).

## 5.5 Further Analysis

### 5.5.1 *Few-shot Learning*.
A significant advantage of pre-trained models is that they can still work well when downstream task datasets are relatively scarce. Many practical applications related to tabular data face the challenge of scarce annotated data, such as medical diagnosis [24]. Therefore, we conducted extensive experiments to explore the practical effectiveness of CM2 in the context of few-shot learning settings. Specifically, for each downstream tabular dataset, we randomly sample 5/10/20 samples from each class to construct three new 5-shot/10-shot/20-shot tabular datasets. We then fine-tune our pre-trained tabular model on these new few-shot datasets. The experimental results are shown in Figure 9, we observe that: **(i)**-Cross-table pretraining has resulted in a significantly enhanced performance compared to learning from scratch (*w/o pretraining*). **(ii)**-Pre-trained model exhibits a greater improvement when the number of samples is less. For example, the boost is most significant in the 5-shot case while relatively weaker in the 20-shot case. We analyze this due to the shareable knowledge learned through pretraining being relatively more valuable when the available training data is less.

### 5.5.2 *More Table Pretraining Objectives*.
We conduct more experiments to compare with other common table pretraining objectives, including supervised contrastive learning (SupCL) and vanilla supervised transfer learning (Vanilla). Also, this can serve as an evaluation of the universality of OpenTabs we have contributed. We make slight modifications to the previous tabular contrastive learning method to adapt it to cross-table scenarios; refer to Appendix A.1 for details. As illustrated in Figure 6(a), we observe that SupCL pretraining can also enhance model performance, whereas Vanilla tends to degrade on downstream tasks. We analyze this as being due to vanilla supervised learning objective, such as the cross-entropy loss, often making the model over-confident and biased towards certain specific tables, which would degrade its general ability in the cross-table setup. While contrastive learning tends to soften the representation clustering effect, as discussed before [41].

### 5.5.3 *Masking Ratio of pMTM*.
Previous research [13] has suggested that a higher masking ratio is required to achieve better performance in *masked image modeling*, whereas a lower masking ratio is sufficient for *masked language modeling*. In this experiment, we further investigate the impact of masking ratio on prompt Masked Table Modeling (pMTM), as shown in the right subplot of Figure 6(b). We found that the model has high performance between 30% and 50%, with an excessively high masking ratio leading to a steep descent. We analyze that tabular data exhibits high information density, where a change in a single feature value can significantly alter the meaning of a sample. So too high a masking ratio will cause the model to have difficulty in learning the correct feature relationships.

### 5.5.4 *Hyperparametric Sensitivity Analysis*.
We analyzed the sensitivity of different hidden dimensions in the transformer and the learning rate. We randomly select some datasets for the experiment and change the hidden dimensions and learning rate. The experimental results are shown in Figure 7. The settings are consistent with Section 5.1.4 except for the corresponding hyperparameters. It can be seen that CM2 is robust to the hyperparameters.

## 6 CONCLUSION

With CM2 and OpenTabs, we hope to initiate the scaled cross-table pretraining for the communities that are related. If perceiving this work through the lens of the development of current LLMs, the scale of CM2 might still be deemed small (50M parameters). Despite that, as we iterate in the paper, we hope to use this work to reach a "BERT moment" for the tabular data mining, where CM2 roughly matches the parameter scale of the base model in the family of BERT [10].

On the bright side, the volume of available tabular data is truly gigantic — wherever a database system is deployed there will be tabular data — but perhaps much more decentralized and abstracted than the text and vision data. In the future, we hope to explore further scaling CM2 and adapting it to more diversified data domains.

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

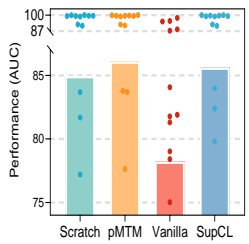 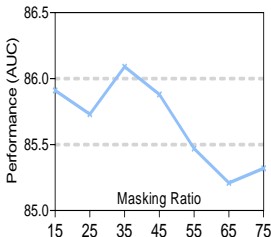

(a) The performance of different table pretraining objectives. Sec. 5.5.2.

(b) Different masking ratio settings in prompt Masked Table Modeling method.

**Figure 6: Analysis experiments on all datasets.**

## A  RELEVANT DETAILS

### A.1  Supervised Contrastive Learning Pretraining Objective

We observe that tabular samples with the same labels tend to have similar feature sets. Based on this observation we make a bold hypothesis: powerful tabular representation should model the invariant factors of feature sets with the same label. We, therefore, propose a *random overlapping subsampling* method to construct positive and negative samples in contrastive learning. In contrast to previous *fixed partitioning* policy [38, 43], our *randomized partitioning* enhances the likelihood of selecting similar feature sets across different tables, further enhancing cross-table pretraining.

Figure 8 illustrates how we randomly sample subsets and divide positive and negative pairs. Specifically, for each row $(\mathbf{x}_i, y_i)$ we randomly sample $k$ feature subsets $\{\mathbf{s}_i^1, \mathbf{s}_i^2, \ldots, \mathbf{s}_i^k\}$ and set all their labels to $y_i$. There will be a partial overlap of features between subsets. In this way, subsets with the same label form positive pairs, and with different labels form negative pairs. Contrastive loss is:

$$\mathcal{L}_{pretrain}^{CL}(\mathbf{X}, \mathbf{y}) = \frac{1}{|B|} \sum_{i \in B} \frac{1}{|P(i)|} \sum_{p \in P(i)} \Psi(\mathbf{z}_i^{CLS}, \mathbf{z}_p^{CLS}), \quad (10)$$

$$\Psi(\mathbf{z}_i^{CLS}, \mathbf{z}_p^{CLS}) = -\log\left(\frac{\exp(sim(\mathbf{z}_i^{CLS}, \mathbf{z}_p^{CLS})/\tau)}{\sum_{i' \in B} \exp(sim(\mathbf{z}_i^{CLS}, \mathbf{z}_{i'}^{CLS})/\tau)}\right), \quad (11)$$

where $B$ is the set of samples in a batch; $P(i) = \{p | p \in B, p \neq i, y_i = y_p\}$. Prior work [38] has introduced that *self-supervised tabular contrastive learning* [3] is applicable to data with a rich diversity of classes, a condition that the majority of tabular data do not meet. Hence, it isn't suitable for our cross-table learning.

## B  DETAILS OF EXPERIMENTS

**Environment.** All experiments are conducted with 8 GPU V100, Intel(R) Xeon(R) Gold 6240 CPU @ 2.60GHz, and 128GB RAM.

**Hyperarameters.** In our experiments, we uses a 3-layer transformer, where the embedding dimension of the token is 128, the hidden dimension of the middle dense layer is 256, and the self-attention module has 8 heads. We use a dropout of 0.15 in all attention layers and feed-forward layers. We choose ReLU for all activation functions. We train CM2 using Adam [22] optimizer with a learning rate in {5e-5, 1e-4, 3e-4}, where the learning rate of

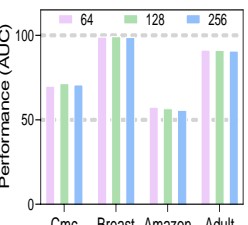 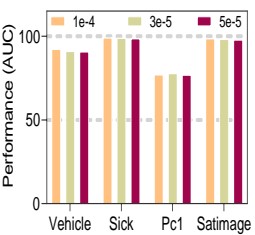

(a) Comparison of results with different hidden dimensions in transformer.

(b) Comparison of results with different learning rate.

**Figure 7: Sensitivity analysis for hyperparameters.**

the fine-tuning phase will be smaller than that of the pretraining phase. Batch size is in {64, 128, 256}. In all other experiments except in Section 5.5.3, we set the masking ratio to 30% for prompt Masked Table Modeling (pMTM) pretraining objective. Within supervised contrastive learning pretraining objective, we set the number of random partitions to 2. In the pretraining phase, we set the maximum training epoch to 500. In the fine-tuning phase, the maximum training epoch is 200 and the patience value is set to 20 for early stopping.

**Baseline Implementation.** The setup of our baseline follows the previous work [43] and includes the following methods:

- **Logistic Regression**: Use the default setting of the package Scikit-Learn. The maximum number of estimators is set to 1000.
- **XGBoost**: Implemented based on the XGBoost package. We set the maximum number of estimators in {50, 100, 300} and the max depth in {5, 8, 10}.
- **LightGBM**: Implemented based on the LightGBM. We set the maximum number of estimators in {50, 100, 300} and the max depth in {5, 8, 10}.
- **MLP**: Dense layers with hidden dimensions {256, 256}. Dropout with a rate of 0.1 is used. They are trained with batch size ∈ {16, 32, 64, 128}, learning rate ∈ {5e-5, 1e-4, 1e-3}, and early stopping patience of 5 with 100 maximum epochs.
- **TabNet**: Use the official implementation with the default recommended parameters[9]. Trained with batch size ∈ {16, 32, 64, 128}, learning rate ∈ {1e-4, 1e-3, 2e-2}, $n_a, n_b \in \{8, 16, 64, 128\}$, $\gamma \in \{1.3, 1.5, 1.8\}$, categorical embedding dimension ∈ {1, 8, 16} and early stopping patience of 5 with 100 maximum epochs.
- **DCN-v2**: Use the implementation by paper [12][10]. The number of cross is 2. The dropout rate for the feedforward component is 0.1. MLP part has two dense layers of dimension {256, 128}. Trained with batch size ∈ {16, 32, 64, 128}, learning rate ∈ {5e-5, 1e-4, 1e-3}, and early stopping patience of 10 in 100 maximum epochs.
- **AutoInt**: Use the implementation by paper [12][10]. The attention layer number is set to 2. The attention head number is set to 2. MLP part has two dense layers of dimension 256, 128; dropout deactivated; trained with batch size ∈ {16, 32, 64, 128}, learning rate ∈ {5e-5, 1e-4, 1e-3}, and early stopping patience of 10 in 100 maximum epochs.

---

[9]https://github.com/dreamquark-ai/tabnet
[10]https://github.com/Yura52/tabular-dl-revisiting-models

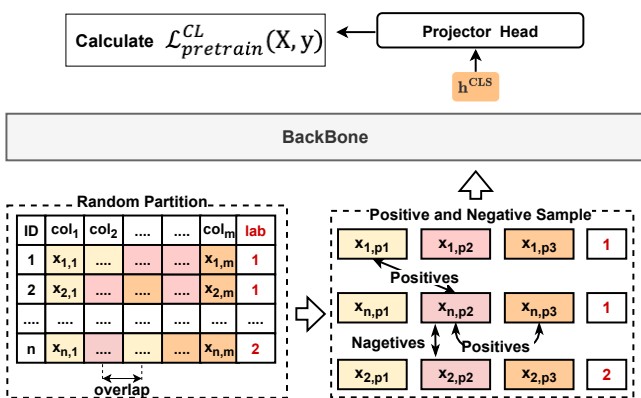

**Figure 8: An illustration of supervised contrastive learning pretraining objective. For each row, we randomly sample K feature subsets (same color means belonging to the same feature subset). Note that there may be overlap among the features of these subsets. The feature subsets with the same label are positive samples, and those with different labels are negative samples.**

- **SAINT**: Use the official implementation[11]. The embedding size is 32 dimensions. 6 transformer layers are used. The number of heads of attention is $\in \{4, 8\}$. The dropout rate is 0.1 in all attention layers and feed-forward layers. Inside the self-attention layer, the q, k, and v vectors are of dimension 16, and in the intersample attention layer, they are of size 64.
- **FT-Transformer**: Use the official implementation[12]. Feed-forward component has 128 dimensions. 2 transformer layers are used. The number of heads of attention is $\in \{2, 4, 8\}$. The dropout rate is 0.1.
- **TransTab**: Use the official implementation[13]. Token embedding has 128 dimensions. 2 transformer layers are used. The number of heads of attention is 8. We train the model on all downstream task data taking batch size 64, learning rate 1e-4, dropout rate 0, and early stopping patience of 10 in 100 maximum epochs. We run the pretraining, transfer learning, and vanilla supervised training methods in the paper, and take the highest score.
- **XTab**: Use the official implementation[14]. We use heavy finetuning in its experiment setup. Specifically, we use an early stopping patience of 3 epochs. The maximum number of epochs is set to infinity. We use his 2000 epoch pre-trained model. Trained with batch size 64.

## C DATASET

### C.1 More Information about OpenTabs

For each table, our data cleaning protocols include, but are not limited to:

(1) *Check the semantic degree of the column names.* For example, the column names {*user_age*, weight, *monthly_income*} have high

---

[11]https://github.com/somepago/saint
[12]https://github.com/Yura52/rtdl
[13]https://github.com/RyanWangZf/transtab
[14]https://github.com/BingzhaoZhu/XTab

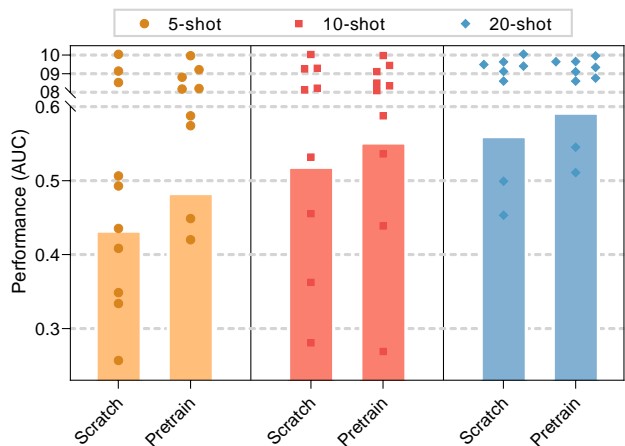

**Figure 9: Few-shot results on all datasets. Scratch represents learning from scratch (*w/o pretraining*).**

semantic information, while the column names {*f1, f2, xyz*} have low. We calculate a cumulative semantic relevance score for each table and, as part of our protocol, exclude tables which less than 50% of the semantic score.

(2) *Check the missing values.* The datasets with more than 40% missing values are discarded. Because too many missing values can easily lead to biased or inaccurate results in the pretraining phase.

(3) *Data Preprocessing.* For categorical features in the tables, we restore them to their original textual values whenever possible. As for numerical features, we employ normalization to mitigate the impact of inconsistent measurement units across different tables (e.g., kilograms vs. grams).

### C.2 Details of the downstream tabular tasks

**Table 5: Statistical Information of Downstream Tabular Tasks**

| Dataset Name | R/C | Samples | Numerical | Categorical | label classes | Source |
|---|---|---|---|---|---|---|
| Breast | C | 699 | 9 | 0 | 2 | https://archive.ics.uci.edu/dataset/15/breast+cancer+wisconsin+original |
| Cmc | C | 1473 | 2 | 7 | 3 | https://archive.ics.uci.edu/dataset/30/contraceptive+method+choice |
| Diabetes | C | 768 | 8 | 0 | 2 | https://openml.org/d/37 |
| Vehicle | C | 846 | 18 | 0 | 4 | https://archive.ics.uci.edu/dataset/149/statlog+vehicle+silhouettes |
| Satimage | C | 6430 | 36 | 0 | 6 | https://archive.ics.uci.edu/dataset/146/statlog+landsat+satellite |
| Sick | C | 3772 | 7 | 22 | 2 | http://archive.ics.uci.edu/dataset/102/thyroid+disease |
| Analcatdata | C | 797 | 0 | 4 | 6 | https://pages.stern.nyu.edu/jsimonof/AnalCatData/Data/ |
| Pc1 | C | 1109 | 21 | 0 | 2 | https://openml.org/d/1068 |
| Adult | C | 48842 | 6 | 8 | 2 | https://archive.ics.uci.edu/dataset/2/adult |
| PhishingWebsites | C | 11055 | 0 | 30 | 2 | https://archive.ics.uci.edu/dataset/327/phishing+websites |
| Cylinder-bands | C | 540 | 18 | 21 | 2 | https://archive.ics.uci.edu/dataset/32/cylinder+bands |
| MiceProtein | C | 1080 | 77 | 4 | 8 | https://archive.ics.uci.edu/dataset/342/mice+protein+expression |
| Car | C | 1728 | 0 | 6 | 4 | https://archive.ics.uci.edu/dataset/19/car+evaluation |
| Segment | C | 2310 | 19 | 0 | 7 | http://archive.ics.uci.edu/dataset/50/image+segmentation |
| Porto-seguro | C | 2000 | 26 | 31 | 2 | https://openml.org/d/44787 |
| Amazon | C | 2000 | 0 | 9 | 2 | https://openml.org/d/44712 |
| Elevators | R | 16599 | 18 | 19 | - | https://openml.org/d/216 |
| Yprop | R | 8885 | 251 | 0 | - | https://openml.org/d/416 |
| Topo | R | 8885 | 266 | 267 | - | https://openml.org/d/422 |
| SAT11 | R | 4440 | 115 | 1 | - | https://www.cs.ubc.ca/labs/algorithms/Projects/SATzilla/ |
| Diamonds | R | 53940 | 6 | 3 | - | https://openml.org/d/42225 |
| House_sales | R | 21613 | 20 | 1 | - | https://openml.org/d/42731 |

