# OpenReview forum: "Towards Cross-Table Masked Pretraining for Web Data Mining"
_ACM.org/TheWebConf/2024/Conference — TheWebConf24_

### Official Review · Reviewer_aot2 · 2023-11-15

**Novelty:** 5
**Technical Quality:** 4

**Review:**

The paper presents a new pre-trained model for tablular data called CM2. In addition, the paper also contributes a set of tabular datasets to-be-used for pre-training and fine-tuning. Overall, the paper is easy to follow and introduces a new method to address tabular data. I do,however, some concerns about the scope of the problem addressed and the analysis provided in the paper. Please see list of Pros and Cons below.


Pros:

S1: The paper is very easy to follow, providing nice illustrations and efficiently utilizing font gestures (bold/underline/...) to highlight the main ideas and insights.

S2: The paper provides open-source resources for the community including code, model and pre-training datasets. This effort also includes a nice benchmarking of tasks (mainly Table 2) that can take us one step closer to Glue[1]-like environment for Tabular data.

S3: The benefits of deep learning (and more specifically pre-trained models based on transformers) are still under-explored for tabular data and I believe that this paper takes this research one step forward.

Cons:

W1: The scope of Tabular Data is unclear:

W1.1: The paper claims to address cross-table training. It is not completly clear to me where "cross-table" comes into play. Spesifically, Section 4.3, which discusses "cross-table" pretraining uses masks over single tables aiming to addess heterogeneity. I think renaming is in-order here. This leads me to cross-table tasks.

W1.2: Not only that the name "cross-table" pretraining here is confusing, also the paper does not introduce real cross-table tasks such as Entity Matching [2], Unionable/Joinable Table Search [3], and others, which are explicit cross-table tasks that can be potentially used for pretraining. I would, at least, expect a discussion about these lines of work in the related work section and have deeply appreciated using these tasks for (pre)training and inference.

W1.3: How does this model address missing data and metadata (e.g., lack of headers)? Specifically, seems like the training of the model depands on the existence of metadata, which seem to be a scarce resource in contemporary environments. [4]

W1.4: The Problem definition is confusing. Based on the definition given in Section 4.1 seems like the tasks being solved are only "row-based", is that really the case (single yi for xi which represents a row)? How does this translate to the cls token?

W2: Some experiment details (+dataset) are vogue and require more details:

W2.1: Some of the improvements seem marginal. It would have been helpful if an additional indication was added to the tables (e.g., stat. sig. test or SD). For example, while the authors claim that "CM2 has an excellent advantage on regression tasks", looking into Table 4 reveals that other than the SAT 11 dataset, all improvements are extremely marginal.

W2.2: I find Section 5.5 too vague and very hard to follow. I do understand that this due to space constraints; however, since it was added I do consider that to be confusing.

W3: Figures are not color-blind inclusive and also cannot be understood in B&W.

Other Comments:

D1: Relevance to track is not declared.

D2: The term "BERT moment" is being used continuously. I have not heard of this term before, is this coined by this paper? If not, I would appreciate a clear explanation in the paper or a reference.

D3: I think adding the avg. sizes of tables to Table 1 can be beneficial.

D4: I find it confusing that the term "web-tables" is commonly used to refer to smaller tables while in this paper it refers to larger tables

[1] https://gluebenchmark.com/
[2] Barlaug, Nils, and Jon Atle Gulla. "Neural networks for entity matching: A survey." ACM Transactions on Knowledge Discovery from Data (TKDD) 15.3 (2021): 1-37.
[3] Fan, Grace, et al. "Table Discovery in Data Lakes: State-of-the-art and Future Directions." Companion of the 2023 International Conference on Management of Data. 2023.
[4] Nargesian, Fatemeh, et al. "Data lake management: challenges and opportunities." Proceedings of the VLDB Endowment 12.12 (2019): 1986-1989.


I acknowledge you reading the rebuttal and responding accordingly.

**Questions:**

See "Review". Specifically W1, W2, D2.

Q1: How do you refer to columns having non-numeric/textual/categorical values (e.g., dates)?

**Reviewer Confidence:**

4: The reviewer is certain that the evaluation is correct and very familiar with the relevant literature

**Scope:**

3: The work is somewhat relevant to the Web and to the track, and is of narrow interest to a sub-community

---

### Official Review · Reviewer_zHj3 · 2023-11-19

**Novelty:** 6
**Technical Quality:** 5

**Review:**

Summary
=======

The paper proposes a large-scale dataset (OpenTabs) and pretrained model for cross-table predictions (CM2), where the model is pretrained on a different set of tables with a different number and order of columns than when making predictions. The dataset contains over 2,000 tables with over 46 million rows in total (2.9 GB of compressed data). The model encodes the tables in a column-invariant way by means of (set) transformers and a new self-supervised training objective.

Pros
====
- Novel large-scale dataset
- Novel large-scale pretrained model for tabular data which is highly relevant for many tasks on the web
- Outperforms previous methods on various downstream tasks (regression, anomaly detection, missing value imputation)
- Source code and data are publicly available


Cons
====
- Some details of the approach could be described in more detail (see below)




Details
=======
- Section 4.1 introduces y_i (classes of labels). However, it is unclear how y_i is used later on. The model is trained by masking and does not require a supervised label. Is the y_i used for the downstream tasks (regression, anomaly detection, missing value imputation)?
- More details about the constructed tabular dataset, such as the sources, types of data included, and any preprocessing steps would be helpful
- "we refine the atomic units “feature” within the table into a sequence of words, which also includes the corresponding column name schema information" How exactly is the column name schema information included?
- How exactly are Sections 4.2.1 and 4.2.2 related? I assume that 4.2.2 describes the encoder. However, it would be great to explicitly state this to make it easier for the reader. By the way, there is some related work on set transformers available.
- "Different from previous tabular reconstruction endeavors [1, 48], our first attempt to use column names as prompt to assist in predicting masked features." This is not a complete sentence and a verb is missing.

**Questions:**

- Can you provide more details on how the tabular dataset was curated? How did you decide which tables to include and which tables to exclude? Can you provide more detailed statistics on the dataset? (exact number of tables by source, number of rows, ...)

**Ethics Review Description:**

no concerns

**Reviewer Confidence:**

3: The reviewer is confident but not certain that the evaluation is correct

**Scope:**

4: The work is relevant to the Web and to the track, and is of broad interest to the community

---

### Official Review · Reviewer_Vefn · 2023-11-22

**Novelty:** 6
**Technical Quality:** 6

**Review:**

This paper addresses the challenges in tabular data pretraining, including proposing a high-quality real-world tabular dataset, an efficient cross-table pretraining framework and a novel pretraining objective, following the prompting and masked ideas from the NLP domain. The pretrained model can be used in various downstream tasks, such as classification, regression and anomaly detection.

As I am not very familiar with tabular data processing, I am not sure whether there are some related works not involved in the paper, besides those appearing in Table 1, especially for the scale issue.

Pros:
1. The pretraining model for tabular data is very significant and worth studying, and the dataset is an important contribution for future research.
2. The model is sound and the experiments are extensive to demonstrate the superiority of this model.
3. The structure as well as the presentation of the paper is fine, and the paper is easy to follow. Core sentences are emphasized to help readers understand the key ideas of the paper.

Cons:
1. The information of Figure 1 is a little insufficient. The challenges of tabular data pretraining are not reflected.
2. There are some typos in the paper. For example, in page 2, line -2, what does "cule" mean? Should it be "cue"?

**Questions:**

There are some typos in the paper. For example, in page 2, line -2, what does "cule" mean? Should it be "cue"?

**Reviewer Confidence:**

2: The reviewer is willing to defend the evaluation, but it is likely that the reviewer did not understand parts of the paper

**Scope:**

3: The work is somewhat relevant to the Web and to the track, and is of narrow interest to a sub-community

---

### Official Review · Reviewer_Z57d · 2023-11-23

**Novelty:** 2
**Technical Quality:** 2

**Review:**

The paper introduces CM2, a novel approach for tabular data analysis, leveraging cross-table pretraining with a focus on feature interaction modeling. CM2 employs a pretraining objective, Prompt Masked Table Modeling (pMTM) to effectively capture structural information in tabular datasets. The paper suggests the potential for further scaling and adaptation to different data domains in the future.

**Questions:**

Strength:
	The introduction of pMTM is an interesting pretraining objective, showcasing the paper’s contribution to capturing prior structural information in tabular data.
	The paper demonstrates performance gains in few-shot learning settings, making it particularly effective in scenarios with limited annotated data. This represents a practical advantage in domains where labeled data is scarce.
	CM2’s tabular representations exhibit remarkable versatility across diverse downstream tasks, including regression, anomaly detection, and missing value imputation. This showcases its adaptability and applicability to a wide range of real-world scenarios.

Weakness:
	While the paper provides an extensive list of baseline models, a more detailed analysis and discussion of their strengths and weakness in the context of tabular data could significantly enhance the paper’s evaluation section. Additionally, absence of widely recognized and popular methods like TaBERT, TUTA, TURL, TAPAS etc. for tabular data learning from baseline comparisons is a notable gap. Including these models in the evaluation would provide a more comprehensive assessment of CM2’s performance and clarify its standing compared to state-of-the-art tabular learning methods.
	The paper mentions tuning the Transformer architecture for the permutation invariance property of tabular data but lacks a detailed discussion of this tuning process. Providing more insights into how the model specifically addresses permutation invariance and the implications of discarding positional encoding would enhance the understanding of the model’s design choices.
	The paper does not extensively justify the choice of a 128-dimensionla embedding for tokens in the transformer architecture, especially when the widely adopted BERT base model employs a 768-dimensional embedding. The absence of an explicit justification raises questions about the trade-offs and potential loss of information when transitioning from the original 768-dimensional space to the chosen 128-dimensional embedding.
	The paper mentions the release of a large pre-trained tabular model (〖CM2〗_V1) trained on 2k datasets. However, the adequacy of this data size for training a model with approximately 50 million parameters is not extensively discussed.

**Ethics Review Description:**

-

**Reviewer Confidence:**

4: The reviewer is certain that the evaluation is correct and very familiar with the relevant literature

**Scope:**

3: The work is somewhat relevant to the Web and to the track, and is of narrow interest to a sub-community

---

### Decision · Program_Chairs · 2024-01-22

**Decision:**

Accept

**Comment:**

The paper addresses the underexplored application of pretraining techniques, specifically focusing on tabular data mining on the web, a domain that has not yet experienced a "BERT moment." The proposed CM2 framework introduces a high-quality real-world tabular dataset, an innovative cross-table pretraining approach, and a novel pretraining objective, pMTM, demonstrating state-of-the-art performance and highlighting the potential of cross-table pretraining for improving downstream tasks.

 - The paper is well written in an under-explored area and provides a solid benchmark as well as a new dataset.
 - Experimental details are unclear with incremental improvements in certain tasks and some lack of clarity in the last sections